# TELEHEALTH and digital health platforms in promoting access to sexual reproductive health self care among youth: A case of rocket health services in Uganda

Vincent Ssenfuka[1,2]*, John Mark Bwanika[1], Louis Henry Kamulegeya[1], Elizabeth Ekirapa Kiracho[2], Martha Akulume[2], Lynn Atuyambe[2,3]

1 Department of Operations and Projects, The Medical Concierge Group, Kampala, Uganda, 2 Department of Health Policy, Planning and Management, Makerere University School of Public Health, Kampala, Uganda, 3 Department of Community Health and Behavioural Sciences, Makerere University College of Health Science, School of Public Health, Kampala, Uganda

* ssenfukavicent@gmail.com

## Abstract

Sexual Reproductive Health (SRH) self-care offers a pathway for low income countries to advance towards Universal Health Coverage by empowering individuals, families, and communities to prioritize their SRH needs independently of healthcare providers. Facilitating access to SRH products is crucial for embracing self-care and digital health technologies hold promise for enhancing accessibility. This study explored the role played by rocket health digital platforms in enhancing accessibility to SRH self-care products among youth in Uganda. Employing a cross-sectional design with a mixed-method approach, the study involved key informant interviews with youth who had purchased SRH self-care products from Rocket Health in 2022, as well as key staff at Rocket Health. Quantitative data were extracted from Rocket Health's Electronic Medical Records covering the period from January 2022 to December 2022.More males (57%) utilized digital platforms for SRH compared to females (43%). The highest utilization was via the E-commerce platform (49%) while the least was via the voice call platforms (4%). A notable portion of youth (30%) still relied on visiting the pharmacy. Contraception products were predominantly consumed through digital platforms (44%), whereas self-testing were less frequently utilized (14%). The study also identified key resources such as the digital infrastructure that maximize the potential of digital health platforms in enhancing SRH self-care. By gaining insights into the digital infrastructure, preferences, barriers, and financial considerations associated with accessing SRH self-care services through digital platforms, targeted interventions such as access to contraceptives, awareness programs, prevention and treatment of Sexual Transmitted Infections can be developed to promote positive SRH outcomes among youth.

of the Creative Commons Attribution License, which permits unrestricted use, distribution, and reproduction in any medium, provided the original author and source are credited.

**Data availability statement:** The data that support the findings of this study are publicly available from Mendeley Data with identifiers doi: https://doi.org/10.17632/sx2g2mx3p6.1 and doi: https://doi.org/10.17632/tfrff4bhdw.2.

**Funding:** The author(s) received no specific funding for this work.

**Competing interests:** The authors have declared that no competing interests exist.

## Author summary

This study examines the role of digital health platforms in promoting sexual and reproductive health (SRH) self-care among youth in Uganda. Platforms such as e-commerce shops, USSD, voice calls, and social media have shown potential to increase access to SRH services by providing convenience, confidentiality, and flexibility. We focused on Rocket Health, a leading telehealth provider in Uganda that integrates these digital platforms to deliver SRH self-care. Using a mixed-methods design, we analyzed service utilization data from electronic medical records and conducted interviews with youth clients and staff. The findings indicate that e-commerce was the most frequently used channel for obtaining SRH products, while teleconsultations and counseling enhanced safe use and built client trust. Contraceptives and sildenafil emerged as the most commonly accessed products, highlighting both opportunities for expanded access and challenges related to potential misuse. Overall, the study demonstrates the promise of digital health in expanding youth-friendly SRH services and underscores the importance of safeguards, such as mandatory prescriptions and integrated health education, to ensure responsible use. These findings provide valuable insights for policymakers and practitioners aiming to scale up safe, equitable digital SRH services in Uganda and similar contexts.

## Introduction

Globally, there could be enough healthcare workers to achieve the Sustainable Development Goals index threshold but due to uneven distribution, there are countries with high needs-based shortage of healthcare workers especially in low-income countries [1]. In Africa, it is estimated that the needs-based shortage of healthcare-workers by 2030 will be 45% [2]. This highlights the necessity for African countries to adopt self-care strategies, empowering individuals, families, and communities to enhance health, prevent disease, maintain well-being, and manage illness and disability, either independently or with minimal healthcare provider support [3]. Youth in Uganda encounter numerous barriers in accessing Sexual and Reproductive Health (SRH) services despite high demand and the unmet demand [4]. Factors such as stigma, cultural taboos, and financial constraints often prevent young people from seeking the care they need [4]. Privacy concerns are also a significant deterrent; many fear that visiting health facilities could expose them to unwanted attention or that their medical information could be shared without their consent [5]. According to the Uganda Demographic and Health Survey [6], the teenage pregnancy rate stands at 25%. This high rate of adolescent fertility is linked to poor access to SRH information, limited availability of youth-friendly services, and deeply rooted socio-cultural norms that discourage open dialogue about sexuality [4]. Given these challenges, there is a growing need for innovative solutions like digital health and self-care platforms, which allow youth to access healthcare independently or with minimal reliance

on healthcare providers [7]. Digital platforms mitigate these concerns by offering confidential, user-controlled environments where individuals can privately access accurate information, consult with qualified clinicians through telehealth services, and order SRH products for discreet home delivery [8]. This model empowers users to manage their health on their own terms, free from the fear of social exposure or judgment. [9,10]. The global strategy on digital health has a vision to improve health for everyone, everywhere by accelerating the development and adoption of appropriate, accessible, affordable, scalable and sustainable person centric digital health solutions [11]. Mobile phone penetration in Uganda is growing rapidly, driven by increased affordability and availability of mobile devices. As of recent reports, Uganda has about 37.3 million mobile phone subscriptions, representing a penetration rate of about 75% of the population [12]. The widespread availability of mobile phones makes them a valuable tool for delivering healthcare services through mobile health (mHealth) platforms, such as telemedicine and health information applications. Internet access is also expanding, though the penetration is lower compared to mobile phones. Uganda's internet penetration rate is estimated at 59%, with approximately 27 million people using the internet [12]. Most users access the internet via mobile phones, and the majority of these users are youth, who are more familiar with technology and more likely to use online platforms [13]. However, internet access remains limited in rural areas, where infrastructure and affordability are key barriers [14]. Rocket Health is a private digital health company established in Uganda in 2012 with the mission of making quality, affordable healthcare accessible to all through the power of technology. Headquartered in Kampala, Rocket Health has positioned itself as a pioneer in Uganda's digital health ecosystem by integrating telemedicine, mobile laboratory services, pharmacy deliveries into a single service model. The company's approach enables users to reach licensed medical professionals remotely, order medications and self-care products via an e-commerce platform. Over the years, Rocket Health has played a critical role in advancing digital health adoption in Uganda, particularly during the COVID-19 pandemic, when it provided uninterrupted access to essential health services amid widespread mobility restrictions.[15,16].

In the context of SRH, Rocket Health offers convenient access to quality SRH services. This service is offered through an in-house 24/7 medical call center staffed with doctors, counselors, and pharmacists that offer remote solutions to clients' inquiries via voice (Hotline) and text (SMS, WhatsApp) platforms. Furthermore, the service includes remote deliveries of pharmacy and laboratory sample pick-ups. In addition, Rocket Health operates an online medical e-Shop (shop.rockethealth.ug) which offers a self-service option for any SRH services. Rocket Health also provides a physical pharmacy where clients can walk in and pick the products they ordered online. This virtual setting offers a more convenient, private, and less stigmatizing space where users can order and receive their SRH services [17]. Follow-up calls and automated SMS reminders are used to collect information on health outcomes and the impact of the health services offered. This approach helps to overcome the stigma associated with SRH care, enabling individuals to seek services they might otherwise avoid [18]. The COVID-19 pandemic further demonstrated the critical role of digital tools in promoting Sexual and Reproductive Health Rights [9]. This study aimed to explore the role played by rocket health digital platforms in promoting access to sexual reproductive health self-care products among youth in Uganda.

### Rocket Health digital platforms

**Voice call-in platforms.** Through customization of the Asterisk software, a voice call-in platform was developed that allows immediate and real time interactions between doctors and patients while meeting all data capture needs. The system was designed with the ability to recognize the caller for all future calls to follow up and also monitor progress for patients on medication. This platform offers clients a way to access a doctor consultation via voice call.

**Short messaging system (SMS) and social media.** Rocket Health provides a platform where prospective clients send a keyword to a short code to receive health information and also speak to a healthcare professional. The messaging platform was integrated with the Rapid pro software which enables the sending of regular automated health content and reminders. Social platforms like WhatsApp, Twitter (X) and Facebook are all combined in an application called "Trengo" where the Rocket Health team can handle client inquiries from all the platforms on one dashboard.

**Unstructured supplementary service data (USSD).** Rocket Health launched a USSD platform in 2021 that allows clients to access health services through dialing a short code (*280#). This means that even people without smartphones and no access to the internet could still access SRH self-care services remotely without moving to a health facility. This short code provides options such as doctor consultations, laboratory and pharmacy services.

**Online e-shop.** Launched in 2019 by Rocket Health to provide an online self-service platform for clients to access health care. The platform was designed to allow clients to log on to a web link and purchase a health care service with ease. The services include doctor consultations, laboratory service and pharmacy services. This is a user-friendly self-service platform where a client is presented with a catalog of products they can order.

## Methods

### Study site

We conducted the study in areas served by Rocket Health, namely Kampala and Wakiso districts. Kampala district is divided into 5 divisions and 76 parishes with a population of 6,709,900 in an area of 8451.9 km$^2$ [19]. Wakiso district is divided into 2 counties, 17 sub counties and 131 parishes with an approximate population of 2,915,200 people in an area of 2704 km$^2$ with youth accounting for 36% [20]. Kampala and Wakiso are predominantly urban with trading as the primary economic backbone for a significant portion of their population.

### Study population

The study population included all the male and female youth (18–30 years) in Kampala and Wakiso districts that had used the rocket health services by purchasing reproductive health self-care products such as sanitary pads, condoms, contraceptive pills, HIV self-test kits and others through digital platforms such as USSD, online E-commerce platform and telephone calls and social media platforms. The study also involved health care providers at rocket health who had participated in the development, deployment and utilization of the digital health platforms in the delivery of sexual reproductive health self-care to youth.

### Study design

We employed a descriptive cross-sectional study design incorporating a concurrent mixed methods approach that gathered both quantitative and qualitative data simultaneously. These datasets were analyzed independently and then integrated to provide a comprehensive understanding of the study.

### Sample size

**Qualitative data (key informant interviews).** We conducted 7 key informant interviews with the rocket health staff and 6 key informant interviews with the youth that have utilized rocket health for SRH self-care products such as self-testing kits, contraception and SRH wellness products.

**Quantitative data.** We extracted all the relevant data directly from the rocket health electronic medical system of all records of clients aged 18–30 who accessed self-care SRH services through digital health platforms in 2022. We utilized filters such as date (Jan 2022 - Dec 2022), age [18–30] and contact type (digital platforms). The data obtained was subjected to the cleaning process before it was imported to Stata analytical software for further analysis. Furthermore, a previous survey conducted in Jan 2022 was used to extract costing data and also triangulate data collected from the interviews.

### Sampling procedure

**Key informant interviews.** Participants for key informant interviews with youth were selected purposively based on their age (youth aged 18–30 years), location (Wakiso or Kampala) and frequency of utilization of the rocket health sexual reproductive self-care products and services. We selected individuals that had utilized the service the most times. These

were obtained from the data extracted from the rocket health electronic medical records system using identification numbers. Participants were further stratified by gender to ensure balanced representation. We ultimately selected 1 male and 2 females from each district with the highest utilization of Rocket Health services.

Selection of staff participants was done purposively based on their knowledge and experience obtained while working at rocket health for at least 2 years. We selected 2 individuals from management, 2 from software development, 1 doctor and 2 field technicians to capture operational and strategic insights related to service delivery.

**Secondary data.** Data was extracted from the rocket health Electronic Medical Records System (EMR). The time period considered for data extraction from the Rocket health EMR was from January 2022 to December 2022. The aim was to obtain data collected after lockdown had been fully lifted in Uganda. This is because lockdown might have influenced people using digital health services. Records were excluded if they belonged to individuals outside the target age range or if the SRH service accessed was part of a bundled non-SRH-related care package, which could confound the analysis. Initially, 1,653 SRH-related client records were identified in the EMR database. After applying inclusion and exclusion criteria and removing incomplete or duplicate entries, a final dataset of 948 eligible records was used for analysis.

**SRH self-care products.** SRH self-care products considered in this study were categorized into the following;

The products purchased were divided into three categories: contraception, self-testing and SRH wellness. Contraception included all products that were used to prevent pregnancy such as condoms, oral and injectable contraceptives. Self-testing included products that gave an individual the capability to perform self check-ups without requiring a health worker such as HIV kits, pregnancy kits, and ovulation kits. SRH wellness included products that were used to enhance the physical, emotional, mental and social well-being in relation to SRH. These included supplements, sildenafil, sanitary pads and lubricants.

**Digital health platforms.** Platforms considered in this study included voice call platform, USSD, `social media, online E commerce platform.

**Resources for service provision.** This includes the inputs included in the Theory of Change which include

i) Healthcare infrastructure, i.e., human resource, laboratory, pharmacy and medical equipment.

ii) Ethical, legal and regulatory infrastructure, i.e., policies in place

iii) Digital infrastructure such as computers, internet, networks, mobile phones

vi) Education, i.e., training resources

## Data collection

Data collection for the key informant interviews was conducted using standardized key informant interview guides respectively. We recruited two experienced research assistants who also had a good working knowledge of English and trained them in data collection for this study. The tools were pretested and adjustments made to ensure scientific rigor. Investigators also actively participated in the data collection process. The Key informant interviews lasted on average one hour and all were audio recorded with consent.

The quantitative method involved secondary data extraction from the Rocket health EMR, and financial reports. This was done by the Rocket Health data officer and variables extracted included age, sex, location, platform used, service accessed, date and cost. This provided data on commonly utilized SRH self-care products, frequency of utilization, commonly utilized digital health channels and costs involved.

## Data collection tools

**Secondary data extraction.** A checklist for data collection was developed prior to data collection to ensure data on all the required variables was obtained. Variables extracted from the Rocket health EMR included demographics such as age

and sex, digital health channels commonly used, i.e., SMS, USSD, Voice, E-commerce platform, SRH self-care products mostly utilized and respective percentages, i.e., condoms, contraception, self-testing kits (HIV and pregnancy).

**Key informant interview guides.** Each interview lasted for approximately 15–30 minutes. The youth interviews were conducted by both phone call and physical interviews and all were in English. The staff interviews were purely physical interviews. All interviews were audio recorded by the interviewers to ensure all conversations were well captured. The youth KII guide explored the different reasons why youth SRH services from Rocket health. Furthermore, it was used to triangulate some data extracted from the Rocket health EMR to provide explanations for the observed data.

The staff KII guide explored the different resources required for running the Rocket health digital health platforms in delivery of SRH self-care products.

## Data management and analysis

**Qualitative data analysis.** Audio recordings were professionally transcribed verbatim. Transcripts were then reviewed for accuracy and familiarity. We began the thematic analysis by first familiarizing ourselves with the data through an initial reading of all the transcripts. Next, we generated initial codes by systematically reviewing the data and identifying key concepts and patterns. The initial codes were reviewed and refined to create final codes that well represented the data. After coding, we developed a codebook by organizing similar codes into categories, which were then reviewed to ensure they accurately represented the data. Following this, we organized the categories into the predefined themes based on the conceptual framework. Summary explanations and quotes were then chosen and presented in the report. Finally, we produced the final report, ensuring that the themes were supported by data extracts and aligned with the research objectives.

**Quantitative data analysis.** Quantitative data extracted from the Rocket health EMR was exported into Microsoft Excel for cleaning. Data was double checked to ensure that it is complete and consistent. STATA/SE 15.0 software was then employed for subsequent analysis. Simple descriptive statistics were used to describe the data. We evaluated the utilization of the different digital health platforms by youth using descriptive statistics (proportions for categorical variables and means for continuous variables) stratified by sex.

i) Costing data

Costing data was summarized into averages such as average expenditure on SRH self-care services by youth. Furthermore, they were also summarized as average expenditure per platform. This was to provide evidence on how much youth spend on different SRH self-care services using digital platforms. This would also provide evidence on which platform were they willing to spend more.

ii) Utilization of digital health platforms

This was summarized into percentages, i.e., percentage of youth that had used a particular digital health platform in a defined period of time.

iii) Commonly utilized SRH self-care products and services

These were summarized into percentages, i.e., percentage utilization for each category (Contraception, self-testing, SRH wellness)

**Ethical review and approval.** Approval was obtained from Makerere School of Public Health Institutional Review Board since the study involved interviewing human subjects.

Approval was sought from the Rocket health organization data protection and privacy authority before accessing their Electronic Medical Records and any other relevant data. A data privacy and confidentiality agreement was signed for access to the data. We then filled in a data requisition form indicating the kind of data needed.

Written and verbal consent was obtained from all subjects before data collection and involvement was voluntary. The consent form introduced the participants to the study objectives and why they should participate.

## Results

### Quantitative results

We obtained 948 records of individual Rocket Health clients aged 18–30 who accessed SRH self-care products from January 2022 to December 2022. The extracted data comprised 57% males and 43% females, with a mean age of 26 years.

### Utilization of digital health platforms

Across the 4 digital health platforms (Voice call, USSD, social media & E-commerce), youth mostly used the online E-commerce platform (49%), followed by the USSD (11%) with social media at 6% and lastly voice at 4%. 30% of the youth accessed SRH self-care products through physical means by walking into the Rocket health pharmacy.

Both males and females mostly utilized the online E-commerce platform to access SRH self-care services (Table 1).

### SRH self-care products consumed by youth via digital health platforms

Across the different SRH self-care product categories, youth mostly accessed contraception (44%) followed by SRH wellness (42%) and the least accessed was self-testing (14%). Among females, contraception was the service mostly accessed (50%). Overall, the most consumed products included emergency contraceptives, sildenafil and sanitary pads. Females mostly ordered for levonorgestrel (emergency contraceptives) which accounted for 22.79% of all SRH self-care products ordered via digital health platforms. Among males, SRH wellness was the service mostly accessed accounting for 49% of the SRH self-care products consumed via digital health platforms. Sildenafil, a drug commonly used to treat erectile dysfunction, was the most consumed product under SRH wellness by males accounting for 20.26% of all SRH self-care products consumed (Table 2).

### Amount of money spent by youth while using digital health platforms for SRH self-care

On average, youth spent 3.2 US dollars for SRH self-care using digital platforms. This included costs such as internet data ($0.13), delivery cost ($1.32) and cost of the product. The internet data cost was considered for the e-commerce platform and social media while the delivery cost was considered for all platforms except the physical means. This is because the USSD and voice call platform did not need internet. Furthermore, the voice call was a toll-free line. On average, males (3.4 US dollars) spent more than females who spent 2.8 US dollars. Youth spent an average of 3.7 US dollars on contraception, 2.8 US dollars on self-testing and 2.8 US dollars on SRH wellness. The time horizon for the costing data was one year. All costing was done for the year 2022 which was the closest complete year at the time of

**Table 1. Utilization of SRH self-care by youth through digital health platforms.**

|  | E-commerce platform | (%) | Physical | (%) | Social media | (%) | USSD | (%) | Voice call | (%) | Total | (%) |
|---|---|---|---|---|---|---|---|---|---|---|---|---|
| **Males** | 300 | 56 | 170 | 32 | 28 | 5 | 27 | 5 | 12 | 2 | 537 | 100 |
| **Females** | 168 | 41 | 113 | 27 | 28 | 7 | 80 | 20 | 22 | 5 | 411 | 100 |
| **Total** | 468 | 49 | 283 | 30 | 56 | 6 | 107 | 11 | 34 | 4 | **948** |  |
| **SRH Category** |  |  |  |  |  |  |  |  |  |  |  |  |
| **Contraception** | 237 | 56 | 108 | 26 | 21 | 5 | 34 | 8 | 21 | 5 | 421 | 100 |
| **Self testing** | 55 | 43 | 39 | 31 | 5 | 7 | 18 | 14 | 7 | 5 | 128 | 100 |
| **SRH wellness** | 176 | 44 | 136 | 34 | 26 | 6.5 | 55 | 14 | 6 | 1.5 | 399 | 100 |

**Table 2. SRH self-care products consumed by youth via digital health platforms.**

| Product name | Number consumed | Percentage % |
|---|---|---|
| Levonorgestrel | 216 | 22.8% |
| Sildenafil | 192 | 20.3% |
| Sanitary pads | 152 | 16.0% |
| Condoms | 127 | 13.4% |
| Pregnancy strips | 87 | 9.2% |
| Mifepristone + misoprostol | 48 | 5.1% |
| HIV self-test kit | 40 | 4.2% |
| Microgynon Fe | 25 | 2.6% |
| La-wash intimate hygiene | 22 | 2.3% |
| Asparagus cinnamon herbal extracts | 8 | 0.8% |
| Lidocaine | 8 | 0.8% |
| Others | 23 | 2.4% |
| Total | 948 | 100.0% |

the study. All costs are at 2022 prices and presented in 2022 USD. Costs in Uganda shillings were translated to USD at a rate of 3,714 (Tables 3 and 4).

## Qualitative results

Thematic analysis revealed the following themes.

### Reasons for utilization of digital health platforms by youth for SRH self-care

The youth had various reasons for using the different platforms including convenience, accessibility, affordability, efficiency and privacy.

The E-commerce platform was the most utilized digital health platform for SRH self-care products and this was due to its easy ordering process and the fact that it's a self-service platform where all products are displayed clearly with prices providing consumers with options. The users noted that the E-commerce platform is very user-friendly since it provides real time product information accessibility, enabling better decision making and reducing duplication of the ordering processes.

*"Um, because it has a catalog of products. I'm just browsing. It's like online shopping really. And I'm so used to online services. So I can easily just browse through and see what I want to order."(Youth KII, 26 yr old female)*

**Table 3. Average amount of money spent per platform (US dollars).**

| | Overall average | Online E- commerce platform | Physical | Social media | USSD | Voice call |
|---|---|---|---|---|---|---|
| Average spend per platform (US dollars) | 3.2 | 4.2 | 1.9 | 3.1 | 2.4 | 2.9 |
| Males (US dollars) | 3.4 | 4.3 | 1.8 | 3.3 | 3.5 | 3.8 |
| Females (US dollars) | 2.8 | 3.9 | 2 | 2.9 | 2.1 | 2.4 |
| CONTRACEPTION (US dollars) | 3.7 | 4.7 | 2.2 | 2.4 | 2.2 | 2.1 |
| SELF TESTING (US dollars) | 2.8 | 3.4 | 1.3 | 3.2 | 3.2 | 4.4 |
| SRH WELLNESS (US dollars) | 2.8 | 3.6 | 1.8 | 3.5 | 2.3 | 3.5 |

**Table 4. Average costs of SRH self-care products accessed through digital health Platforms.**

| Product name | Average cost (US dollars) |
|---|---|
| Levonorgestrel | 2.4 |
| Sildenafil | 2.3 |
| Sanitary pads | 1.7 |
| Condoms | 3.5 |
| Pregnancy strips | 1.2 |
| Mifepristone + misoprostol | 11.3 |
| HIV self-test kit | 6.3 |
| Microgynon Fe | 1.9 |
| La-wash intimate hygiene | 3.0 |
| Depo Provera injection | 1.5 |
| Lubricant | 7.2 |
| Desogestre ethinylestradiol | 4.5 |
| IUD cut380 | 0.4 |
| Ovulation strips | 1.3 |

Another participant mentioned:

*"Maybe because, uh, Rocket health, you can see most of the things they have there. It doesn't need to ask, you can see all the products there" (Youth KII, 24 yr old male)*

Convenience was the primary reason youth used digital platforms for SRH self-care. Most participants stated that this model enables them to maintain their routine activities while addressing their SRH self-care needs. Additionally, it allows them to access SRH self-care services at any time, accommodating their busy and unpredictable schedules.

*"It's really the convenience, mostly, because anywhere you are, you can easily access healthcare, you can easily access information, so it's super convenient. Yeah, and then now when it comes to sexual and reproductive health services, it's convenient, but it also offers you privacy, because the ordering is just between you and your phone".(Youth KII, 26 yr old female)*

Participants also noted that digital health platforms offer them privacy when ordering SRH self-care products and this eliminates the stigma that comes with such products in communities. Participants also noted that digital health platforms provide the ability to seek information without revealing one's identity and this encourages them to actually obtain these products.

*"It's a private thing, and if there's an opportunity to keep it that way, like just keep it between me and that person delivering, like I don't even know that person delivering, I would take that option than if I have to go and have to face, uh, other…., people don't want to be judged.*

*Hmm. Or they are traitors". (Youth KII, 29 yr old female)*

Participants noted that Rocket Health has prices similar to the market prices and this ensures that their pricing remains competitive and reflected the prevailing economic conditions.

*"Um, the price depends on the market price. Hmm. So the, the thing with Rocket Health, they sell on the market prices that I would say they're affordable." (Youth KII,)*

 

**Resources available to effective operation of the Rocket Health digital intervention**

**Digital healthcare infrastructure that support SRH self-care.** Digital infrastructure included computer systems, networks, servers, Application Programming Interfaces (APIs) and databases. Most systems used were developed in-house and this enabled flexibility and scalability. The servers that store the databases and source codes were hybrid (both in-house and external) and this ensured data security and backup for client's information. All platforms, i.e., USSD, Voice call, social media and E-commerce shop are connected to the systems such as the Electronic Medical Records through the APIs. This ensures system interoperability.

*"Yeah. All those tools are built in house here by the software development team. Yeah. And they're all communicating. Yes. So that's the beauty of having the custom-made software. And we can easily modify it depending on the use case" (IT staff)*

The facility operates a technology department tasked with maintaining system functionality and overseeing software development and upgrades. The software development process initiates with various teams generating requirements, which are subsequently submitted to the technology department for development. Retention of staff posed significant challenges within the technology department, exacerbating issues during system downtimes. This was primarily due to systems being developed by individuals who subsequently departed the company.

*"We have software developers that are in house that are able to develop the systems that we have. Over 90% I can say over 95% of all the systems that we use, were developed in house meaning that we sat down, thought of requirements like What do we want the system to do, what kind of system do we want, and what we want it to do. And then we shared that with our tech team and were able to develop something for us" (Operations staff)*

Ethical, Legal and regulatory infrastructure that support digital SRH self-care
All staff participants acknowledged the presence of ethical policies within the company's framework. Additionally, integrity and a commitment to ethics are core company values, ensuring that all staff are dedicated to upholding these principles. All staff are required to sign confidentiality agreements every year to ensure patient confidentiality and company data protection.

*"But every point all our staff when they join in, they sign confidentiality agreements. So we hold them accountable for the data". (Operations staff)*

The laboratory, clinic and pharmacy all have operating licenses and these are renewed each year to ensure compliance. Furthermore, all medical personnel are required to register with the relevant professional bodies before they can start their work.

*"Definitely. Because, uh, field people, they are nurses, lab technicians, or pharmacy technicians. They're all registered medical personnel with valid practicing licenses of that particular year. Yes. So, uh, in the medical profession, uh, that particular individual has to be licensed. Of course, the facility license for premises, but also the individuals. The individuals have to be licensed. You cannot work without being licensed". (Field Agent 2)*

**Education and training resources**

The staff acknowledged the existence of a quality assurance team that was responsible for quality monitoring and training of all staff. All interactions with clients from the different platforms were recorded and stored in a specific database that is accessed by the QA team. The QA team uses this data to identify training needs and then organize training sessions

tailored to the identified needs. These include foundation soft skills, phone call etiquette, problem solving and others. These internal training coupled with external training ensure that the team is well accustomed to the changing trends in technology.

*"So all these calls are recorded. And then we have a team, which we call the QA team, which uses the QA system. So these people log into that system, that's where they have access to all those call recordings which they can listen to. And then you know.. do whatever they need to do to ensure that quality is maintained". (IT staff)*

### Healthcare infrastructure that supports digital health

The facility had a skilled workforce of medical doctors, laboratory technicians, nurses and pharmacy technicians who work in shifts to ensure full coverage. A shift normally consisted of seven medical doctors, three in-house laboratory technicians, four in-house pharmacy technicians and 15 field technicians. All these were well trained in the different systems used in the delivery of services to clients. This was essential in providing quality health care to the youth seeking SRH self-care.

*"We actually have medical doctors, well trained. So again, very expensive in terms of human resources. Yes. The lab is run by lab techs. Again, pharmacy run by pharm techs. Qualified Pharm tech, the clinics by qualified nurses". (Operations staff)*

Field staff were provided with gadgets such as motorbikes, smart phones, protective gears and other devices used in provision of remote healthcare to ensure high quality service to clients.

*"I need a working laptop. Mm-hmm. Yeah. Working laptop, of course. Fully charged. Mm. I need my phone, my smartphone fully charged. I need a MiFi with data. Mm-hmm. Also fully charged. Uh, I need a functional motorcycle with enough fuel. Then I need protective gears" (Field staff)*

This service was able to reach clients in Kampala and Wakiso districts within 45 km radius. For the clients who were beyond the distance, a courier service from an external company was used to deliver client SRH self-care products.

*"Our coverage….we are able to work on clients who are within a 40 to 45 kilometer radius from where we are. So that's wide it means Entebbe, We go, Okay, we go to quite far places in Kakiri, Nabbingo, Nsangi and all that we're able to reach" (Operations staff)*

## Discussion

This study aimed to conduct a formative evaluation of the rocket health digital platforms in promoting access to sexual reproductive health self-care products among youth in Uganda so as to provide evidence about the strengths of digital health technologies in the realization of the access framework in sexual reproductive health. Our discussion highlights the different findings and their implications.

### Utilization of digital health platforms

The study revealed varying patterns in the utilization of Rocket Health digital platforms among youth in Uganda with the online E-commerce platform emerging as the most utilized channel. The study also indicated that more males utilized digital platforms for SRH compared to females which is in agreement with a study by [21] which indicated that digital

technology and mobile applications are an integral part of men's lives. This suggests a strong preference for online platforms among youth, possibly due to their convenience and accessibility.

Interestingly, both male and female youth predominantly utilized the online E-commerce platform to access SRH self-care services. This finding indicates the importance of gender-neutral approaches in digital health interventions, ensuring that both sexes have equal access to essential SRH self-care resources [22]. The convenience and widespread access to mobile and internet services, especially in urban areas, make e-commerce a highly utilized channel. This aligns with Levesque's concept that accessible services encourage higher utilization by removing physical barriers to care [23].

The USSD platform, used by 11% of youth, represents another significant mode of access. USSD services are often used due to their simplicity and accessibility on basic mobile phones without requiring internet connectivity [24]. This makes USSD a valuable tool in regions with limited internet access or among populations without smartphones [25]. Although social media platforms are widely popular for social interactions and sharing information, they accounted for only 6% of SRH self-care product access in this study. This low percentage suggests that while youth may turn to social media for information and support, they tend to prefer more direct and secure methods, like e-commerce, when it comes to purchasing products. This finding aligns with a previous study by [26], which showed that most youth do not rely on social media for health-related information. The USSD and social media platforms registered low usage compared to the e-commerce platform, indicating the significance of having a comprehensive product catalog on digital health platforms. Many respondents expressed a preference for a platform where they could independently order SRH self-care products without assistance from a third party. Interestingly, some respondents indicated a preference for a mobile application as an alternative option.

The voice call platform emerged as the least utilized among the digital health platforms that were available at Rocket health. This was due to concerns about privacy and confidentiality, particularly for sensitive SRH-related discussions as most youth are not comfortable with such discussions [27]. Most respondents preferred going online and ordering for their products in privacy without having to speak to someone. Furthermore, the voice platform offered only two options, i.e., English and Luganda and this was insufficient to accommodate all youths' languages, leaving other languages unrepresented. This demonstrates the significance of considering cultural and linguistic perspectives in the design and the delivery of digital health technologies [28].

It's noteworthy that while digital platforms play a significant role in accessing SRH self-care services, a considerable portion of youth (30%) still relied on physical means by walking into pharmacies. This finding aligns with a study conducted in four Central European countries (Czech Republic, Hungary, Poland, and Slovakia), which found that most individuals still preferred to purchase medicines from community pharmacies and regarded them as the most trustworthy source of medication [29]. This highlights the continued importance of traditional healthcare delivery channels and the need for a multi-faceted approach to address SRH access barriers comprehensively.

The dominance of online E-commerce platforms suggests that implementers of digital interventions should prioritize the development and optimization of user-friendly and secure e-commerce platforms for delivering SRH self-care services [18]. This includes ensuring seamless navigation, privacy protection, and integration with existing healthcare systems.

### SRH self-care products consumed by youth via digital health platforms

Contraception was overwhelmingly the most accessed service, with 50% of females utilizing these products. This finding resonates with a cross sectional study that examined Colombian youth's internet usage to access SRH services [30].

This finding also indicates the critical role of contraception in female SRH care and highlights the need for accessible and comprehensive contraceptive services for young women [31]. Emergency contraceptives were especially popular among women, comprising 22.79% of all SRH self-care products ordered through digital health platforms. This result aligns with a study by [32], which found significant interest in emergency contraceptives among South African women. This is because pregnancy primarily affects women, who therefore seek to prevent it. Several factors contribute to this

trend. Primarily, the need to prevent unintended pregnancies is a major concern for female youth, driving the high demand for emergency contraceptives however general abuse of over the counter procured emergency contraceptives has also been reported [33]. The availability of these products through digital health platforms offers a discreet and accessible solution, which is particularly appealing in contexts where women may face barriers to accessing traditional healthcare services [17]. However, this ease of access can also create a risk for misuse and abuse by youth, who may lack the necessary guidance for proper usage [18,33]. Without adequate counseling, there is a potential for inappropriate or frequent use of contraceptives, which may expose young users to adverse side effects and health complications [34]. Therefore, while digital health platforms expand access to essential SRH products, it is crucial to embed robust educational and counseling services to promote responsible use and safeguard against unintended health risks. Among males, SRH wellness emerged as the most accessed category accounting for 49% of the products consumed. Notably, sildenafil, a medication commonly used to treat erectile dysfunction (ED), was the most consumed product by males. This high consumption of sildenafil highlights several important aspects of men's health and behavior regarding accessing sexual health products through digital platforms. Firstly, it is unclear whether the increased uptake reflects a genuine rise in the prevalence of ED among young men, improved social acceptance of seeking treatment for sexual performance issues, or the influence of targeted marketing by digital health providers. Importantly, existing literature and anecdotal evidence suggest that a significant proportion of young men may use sildenafil recreationally, rather than for clinically diagnosed [35]. This distinction is critical, as recreational use without proper diagnosis may not only mask underlying psychological or health-related causes of sexual dysfunction but also discourage users from seeking comprehensive medical evaluation and long-term care. The convenience and privacy afforded by digital platforms, while beneficial in reducing stigma and improving access, can inadvertently contribute to the misuse of medications like sildenafil [36]. This indicates the importance of integrating stronger digital safeguards such as mandatory teleconsultations before dispensing medication and embedding health education components into digital ordering platforms to promote responsible use and guide clients toward holistic care pathways [37]. However, these safeguards were notably absent from the e-commerce platform, the most frequently utilized channel, raising concerns about unregulated access and the potential for misuse.

Another critical safeguard would be the requirement of mandatory prescriptions for products such as sildenafil and hormonal contraceptives. By ensuring that these are recommended by qualified healthcare practitioners, digital health providers can help prevent misuse, reduce associated health risks, and strengthen the overall accountability of digital SRH service delivery.Self-testing products were the least accessed products via digital health platforms, with only 14% accessing the products. The low usage can be attributed to the limited knowledge about self-testing as indicated by some respondents who expressed unfamiliarity with how to use the self-testing kits. Furthermore, this can also be attributed to a gap in health education regarding the benefits of self-esting [38]. This is also due to the fact that Africans generally do not have a culture of taking up wellness checks such as self-testing [39]. This finding is consistent with a global values and preference survey by [40] which also highlighted low knowledge and uptake for self-testing. The low uptake of self-testing further indicates the significant unmet information needs which is key to advancing self-care in Uganda.

### Costs incurred by youth while accessing SRH self-care services via digital health platforms

The study also analyzed the average expenditure of youth on SRH self-care services using digital platforms. On average, youth spent $3.2 for SRH self-care, with males spending slightly more ($3.4) compared to females ($2.8) on average. This gender difference in expenditure may reflect variations in purchasing behavior, access to financial resources, or preferences for specific products or services. This is because males may have more financial autonomy or face different SRH challenges that lead to higher spending.

The findings indicated that the average cost of accessing SRH self-care services was higher than the daily wages of a lowest paid government worker (LPGW) at $1.7 [41]. According to WHO guidelines, a product is considered affordable if it doesn't exceed the day's income of the lowest paid government worker. The most frequently used service, contraception,

has an average cost of $3.70, suggesting it may be unaffordable for many young people especially the low income youth in rural settings. This aligns with a study by [42], which found that most sexual and reproductive health (SRH) commodities in Uganda's private sector were priced higher than a day's wage.

These findings provide valuable insights into the financial implications of accessing SRH services through digital platforms and highlight the importance of affordability in promoting equitable access to essential SRH resources. Efforts should be made to address affordability barriers to ensure that all youth, regardless of gender or socio-economic status, can access SRH self-care services through digital platforms [43]. Addressing this issue requires concerted efforts from policymakers, healthcare providers, and stakeholders to reduce the cost of SRH services and enhance affordability. This may involve exploring innovative pricing models, subsidies, or incentives to reduce the financial burden on users and improve access to essential SRH resources [42]. The government could also consider providing these products to digital health service providers like Rocket health to increase access.

### Resources available for effective operation of the rocket health digital health intervention

We used a theory of change framework developed by [44] to describe the potential of digital health platforms in advancing SRH self-care to improve sexual health and wellbeing. We examined the resources available as inputs to this framework and how they enable provision of SRH self-care services to youth.

The digital infrastructure enabled remote service provision such as doctor consultations via voice calls and SRH self-care products delivery. It also enabled more access to SRH information using SMS and the online E-commerce platform. Computer systems and networks enable real-time data exchange between the users and providers which facilitate prompt order placement and delivery [8]. Data capture, storage and retrieval was enabled by databases connected to the different systems used by the health care providers. Existence of APIs enable system interoperability and this means that all systems are able to share, retrieve and receive data from other systems [45].

This is a big advantage of inbuilt systems over "off the shelf" systems [46]. Furthermore, these systems were very flexible and could be modified depending on the user needs and advances in technology. However, the downside of this is the fact that individuals who develop these systems can never be retained and challenges occur when such individuals leave [46]. This is because the remaining team may struggle to maintain or update these systems due to a lack of understanding of the original design. Furthermore, system downtimes become more problematic when knowledgeable staff are unavailable to quickly resolve issues. This can lead to extended periods of system inactivity, affecting overall productivity and efficiency. These are critical issues to think about when deciding the kind of systems to employ for sustainability and scalability.

Apart from the technology, presence of a licensed human resource responsible for operating the systems and providing remote SRH self-care services provides a seamless service with convenience and confidentiality to the clients. Licensure is an important aspect of ethics and regulations within the health sector and the same applies to digital health [47]. Human resource development and capacity building is critical when it comes to digital health due to its novelty and the changing trends in technology and this should be enabled by the quality assurance team [18,47].

The digital infrastructure supporting Rocket Health's SRH services aligns with the Levesque framework dimension of accessibility. The presence of systems such as voice calls, SMS, and e-commerce platforms makes SRH self-care services more approachable and accessible to youth. However, as the Levesque framework emphasizes, the sustainability of these services depends on maintaining both the technical and human resources required to support the platforms [23]. Challenges related to staff retention and system downtimes, as highlighted in the discussion, indicate the importance of continuous resource development and system flexibility to ensure that healthcare remains accessible.

### Alignment with the Uganda health information and digital health strategic plan

This study leveraged data from Rocket Health's Electronic Medical Records system, showcasing how digital health service delivery can facilitate the availability of health information for research and informed decision-making. This aligns with the mission of Uganda's Health Information and Digital Health Strategic Plan [13], which emphasizes the use of

data-driven insights to guide decision-making and policy formulation while optimizing health service delivery through digital technologies.

### Digital health platforms as tools for SRH education and counseling

Rocket Health digital platform integrate health information into their service delivery models through social media, offering users credible, age-appropriate, and culturally sensitive content on topics such as contraception, sexually transmitted infections (STIs), menstrual health, and fertility management. This educational function is particularly important in settings like Uganda, where formal SRH education is often limited or inconsistently delivered in schools, and misinformation is widespread [48]. Through teleconsultations and social media content, digital health platforms can correct misconceptions, promote health-seeking behavior, and empower youth with the knowledge needed to make informed decisions about their bodies [4]. By combining product delivery with continuous access to expert information, digital health platforms foster responsible health practices, reduce misuse of SRH products such as emergency contraceptives, and promote long-term engagement with preventive health behaviors. As such, these platforms extend beyond service delivery to become enablers of health literacy and autonomy among youth and other vulnerable populations.

### Study limitations

Our study has some limitations. Firstly, we used a small purposive sample during the key informant interviews, which may not have been sufficient to achieve data saturation, and the participants may not represent the entire population segment studied. Additionally, we did not include participants from diverse educational levels and cultural backgrounds, which could have influenced the depth and variety of the data. This is because we used secondary data that was obtained from electronic medical records and thus some demographic factors like level of education, employment status and socioeconomic status were not obtained. The absence of rural participants may affect the generalizability of the findings to more geographically diverse populations. However, the inclusion of highly educated participants who could articulate the information on SRH self-care provided valuable insights into this under-explored topic.

Recall bias and social desirability bias. We carried out key informant interviews with youth about Rocket health services they had consumed. This poses a risk of possible recall bias since they might not remember some of the information. Furthermore, youth might not reveal the actual truth about the SRH products consumed due to the obvious reasons of stigma related to SRH.

The cross-sectional nature of the study limits the ability to infer causality or track changes in behavior over time. Future longitudinal studies are recommended to assess the sustained impact of digital health platforms on SRH outcomes and behavior change among youth.

The researcher's affiliation with Rocket Health, the organization under study, may have introduced potential bias in responses during staff Key Informant Interviews. To mitigate this, external data collectors were engaged to conduct the interviews independently. Additionally, the analysis was carried out under the close supervision of experienced research supervisors, ensuring adherence to rigorous standards and minimizing any potential bias.

## Conclusion

In conclusion, while digital platforms provide a convenient and accessible means for obtaining SRH self-care services, it is crucial to recognize and address disparities in their utilization. The high usage of the e-commerce platform highlights the need for further exploration on how to enhance this platform for SRH self-care.

The high proportion of emergency contraceptive orders indicates a strong need among young women to manage their reproductive health proactively but also could be an indication of unregulated access to these products. The study indicates that SRH wellness is a primary concern for males, with sildenafil being the most consumed product. This could indicate the unregulated access to this drug and highlights the role of digital health platforms in facilitating access to these crucial health products.

The average expenditure of youth on SRH self-care services through digital platforms, with a slight gender disparity, illustrates the need to address both affordability and gender-specific needs in SRH services.

While digital health platforms provide a great opportunity to advance SRH self-care in Uganda, it is critical to understand the different moving parts of the digital health model to ensure equitable access to SRH care. Our analysis revealed that robust digital infrastructure is essential in enabling remote service provision, including doctor consultations via voice calls and delivery of SRH self-care products.

## Recommendations

Basing on the study findings and conclusions about digital health platforms and SRH self-care, we recommend conducting a cost-effectiveness analysis to compare the expenses and benefits of utilizing digital health platforms for sexual and reproductive health (SRH) self-care with those of traditional physical methods for obtaining SRH self-care products. This analysis should aim to inform policy and practice by providing evidence-based insights to guide policy makers, healthcare providers, and stakeholders in optimizing SRH self-care strategies.

## Acknowledgments

We acknowledge the generous support from MakSPH during the manuscript preparation. Special thanks to the Rocket Health team for the secondary data provided and the insights during the interviews. I would like to also thank the research assistants and the participants for their trust and participation.

## Author contributions

**Conceptualization:** Vincent Ssenfuka, John Mark Bwanika, Louis Henry Kamulegeya, Elizabeth Ekirapa Kiracho.

**Data curation:** Vincent Ssenfuka.

**Formal analysis:** Vincent Ssenfuka, Martha Akulume.

**Funding acquisition:** Vincent Ssenfuka.

**Investigation:** Vincent Ssenfuka.

**Methodology:** Vincent Ssenfuka, Elizabeth Ekirapa Kiracho.

**Project administration:** Vincent Ssenfuka, John Mark Bwanika, Martha Akulume.

**Resources:** Vincent Ssenfuka.

**Software:** Vincent Ssenfuka.

**Supervision:** John Mark Bwanika, Elizabeth Ekirapa Kiracho, Lynn Atuyambe.

**Validation:** Vincent Ssenfuka, Elizabeth Ekirapa Kiracho.

**Visualization:** Vincent Ssenfuka.

**Writing – original draft:** Vincent Ssenfuka, Elizabeth Ekirapa Kiracho.

**Writing – review & editing:** Vincent Ssenfuka, John Mark Bwanika, Louis Henry Kamulegeya, Elizabeth Ekirapa Kiracho, Martha Akulume.

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
