## [Decision Letter · Decision Letter 0]

2 May 2025

Response to Reviewers
Revised Manuscript with Track Changes
Manuscript
**Additional Editor Comments (if provided):**
**Reviewers' Comments:**

**Comments to the Author**

1. Does this manuscript meet PLOS Digital Health’s publication criteria?

Reviewer #1: Yes

Reviewer #2: Yes

2. Has the statistical analysis been performed appropriately and rigorously?

Reviewer #1: Yes

Reviewer #2: Yes

3. Have the authors made all data underlying the findings in their manuscript fully available (please refer to the Data Availability Statement at the start of the manuscript PDF file)?

Reviewer #1: Yes

Reviewer #2: Yes

4. Is the manuscript presented in an intelligible fashion and written in standard English?

Reviewer #1: Yes

Reviewer #2: Yes

Reviewer #1: 1. The abstract follows a logical flow, clearly presenting the background, methods, results, and conclusions of Telehealth and Digital health platforms.

2. The findings offer actionable insights into digital health platforms, which can inform policymakers and healthcare providers on improving SRH access.

3. The study could benefit from a comparison with other digital health platforms or similar initiatives in different regions to contextualize its findings.

4. The abstract presents several percentages and numbers, but there is no indication of statistical significance or confidence intervals. While not always necessary in an abstract, mentioning key trends or significant differences could add depth.

5. The conclusion mentions “targeted interventions” but does not outline what these interventions might be (e.g., subsidy programs, awareness campaigns, or digital literacy training). A brief mention would strengthen the impact.

6. The article focuses more on digital health platforms and their usage rather than on telehealth services that support patients in need. Incorporating telehealth services would strengthen the article.

7. The statement that sildenafil is the most consumed product suggests a significant demand for erectile dysfunction (ED) treatments, but it lacks context.

Is this increase due to an actual rise in ED prevalence among young men, greater social acceptance of seeking treatment, or aggressive marketing by digital platforms?

8. Additionally, some young men use sildenafil recreationally rather than for diagnosed ED. This distinction is important for interpreting the data accurately.

9. This should be included as a downside of easy access to these medications on online platforms, as it may bypass addressing the root cause and seeking proper medical attention.

10. The review acknowledges that digital platforms facilitate access to SRH products, but it does not explore whether they also play a role in education, counseling, or misuse prevention.

Understanding how these platforms influence user behavior beyond simple access would provide a more nuanced view.

11. Expand the discussion to consider how digital health platforms contribute to education, counseling, and responsible use of SRH products.

12. The study addresses a critical public health issue—SRH self-care among youth in Uganda—aligning with global efforts toward Universal Health Coverage.

Policymakers should ensure easy access for all in the digital age while also preventing misuse and the unregulated availability of medications with potential side effects without a physician’s prescription.

Reviewer #2: Revised Manuscript Review: "Telehealth and digital health platforms in promoting access to sexual reproductive health self-care among youth: A case of Rocket Health services in Uganda"

Overall Assessment:

This manuscript explores the vital intersection of public health and technology, specifically the role of digital health platforms in enhancing access to sexual reproductive health (SRH) self-care among Ugandan youth. Given the significant challenges in accessing SRH services in sub-Saharan Africa, the study's context is highly relevant. However, the current draft requires substantial revisions to fully realize its potential. The analysis lacks depth, the discussion needs more engagement with existing literature, and the presentation of findings could be significantly improved.

Specific Recommendations:

Introduction:

Focus on Specific SRH Challenges: Clearly identify prevalent issues facing Ugandan youth, such as teenage pregnancy rates and unmet contraceptive needs. This will strengthen the rationale for the study.

Justify Digital Health's Role: Move beyond general statements about digital health potential. Analyze how these platforms can specifically address identified challenges, such as stigma and confidentiality concerns, and link mobile/internet penetration to the feasibility of interventions.

Contextualize Rocket Health: Provide background on Rocket Health’s establishment, mission, and contributions to Uganda's digital health landscape to integrate it smoothly into the narrative.

Methods:

Detail Mixed-Methods Integration: Clearly explain how qualitative and quantitative data were integrated and how they informed each other. Justify the choice of a cross-sectional design and discuss its limitations regarding causality.

Refine Sampling Strategy: Offer a detailed explanation of the purposive sampling strategy for qualitative data, including criteria for participant selection. For quantitative data, specify inclusion/exclusion criteria and the total number of records in the EMR system before filtering.

Enhance Data Collection Details: Provide information on the development and content of interview guides, including whether they were pilot-tested. Describe the data extraction process from the EMR, specifying the variables collected.

Specify Analytical Techniques: Clarify the qualitative analysis steps, including software used and how themes were identified. For quantitative analysis, detail the statistical tests applied.

Results:

Improve Clarity and Presentation: Restructure the results section for better readability. Use tables and figures effectively to highlight key findings, ensuring all visuals have clear titles and labels.

Deepen Qualitative Presentation: Provide richer descriptions of qualitative findings supported by illustrative quotes. Connect these findings to quantitative results to show their complementary nature.

Clarify Costing Data: Offer a step-by-step explanation of how average expenditure was calculated, contextualizing these costs within the average income or cost of living in Uganda.

Discussion:

Engage with Existing Literature: Move beyond summarizing findings. Critically compare your results with other studies on digital health interventions in similar contexts and discuss the theoretical implications.

Expand Limitations Discussion: Provide a comprehensive discussion of limitations, including the impact of the cross-sectional design on causal inference and potential biases in self-reported data.

Suggest Future Research Directions: Based on findings and limitations, recommend specific future research avenues, such as longitudinal studies to assess long-term impacts.

Overall Writing and Style:

Enhance Clarity and Conciseness: Improve writing clarity by minimizing jargon and using shorter sentences. Focus on logical organization to enhance the overall flow.

Standardize Referencing: Ensure consistent and accurate referencing throughout the manuscript, adhering to PLOS Digital Health's guidelines.

Thank you for considering my feedback.

**Do you want your identity to be public for this peer review?** For information about this choice, including consent withdrawal, please see our Privacy Policy

Reviewer #1: **Yes: ** Bala Nimmana

Reviewer #2: **Yes: ** Taiwo Hassanat Bawa-Muhammad

**Figure resubmission:****Reproducibility:** To enhance the reproducibility of your results, we recommend that authors of applicable studies deposit laboratory protocols in protocols.io, where a protocol can be assigned its own identifier (DOI) such that it can be cited independently in the future. Additionally, PLOS ONE offers an option to publish peer-reviewed clinical study protocols. Read more information on sharing protocols at https://plos.org/protocols?utm_medium=editorial-email&utm_source=authorletters&utm_campaign=protocols

---

## [Decision Letter · Decision Letter 1]

11 Aug 2025

Response to Reviewers
Revised Manuscript with Track Changes
Manuscript
**Journal Requirements:**
**Additional Editor Comments (if provided):**
**Reviewers' Comments:**

**Comments to the Author**

Reviewer #1: All comments have been addressed

publication criteria?

Reviewer #1: Yes

3. Has the statistical analysis been performed appropriately and rigorously?

Reviewer #1: Yes

4. Have the authors made all data underlying the findings in their manuscript fully available (please refer to the Data Availability Statement at the start of the manuscript PDF file)?

Reviewer #1: Yes

5. Is the manuscript presented in an intelligible fashion and written in standard English?

Reviewer #1: Yes

Reviewer #1: The addition of Rocket Health’s background improves clarity and aligns the platform with national digital health efforts.

The expanded discussion on sildenafil use appropriately addresses concerns about misuse but would benefit from specifying if any provider safeguards are in place.

The costing analysis is clearer and more policy-relevant; however, affordability in rural or lower-income youth is not sufficiently addressed.

Limitations are well detailed but should mention the exclusion of rural users, which affects the generalizability of findings.

**Do you want your identity to be public for this peer review?** For information about this choice, including consent withdrawal, please see our Privacy Policy

Reviewer #1: **Yes: ** Bala Nimmana

**Figure resubmission:****Reproducibility:** To enhance the reproducibility of your results, we recommend that authors of applicable studies deposit laboratory protocols in protocols.io, where a protocol can be assigned its own identifier (DOI) such that it can be cited independently in the future. Additionally, PLOS ONE offers an option to publish peer-reviewed clinical study protocols. Read more information on sharing protocols at https://plos.org/protocols?utm_medium=editorial-email&utm_source=authorletters&utm_campaign=protocols

---

## [Decision Letter · Decision Letter 2]

23 Sep 2025

Telehealth and digital health platforms in promoting access to sexual reproductive health self care among youth: A case of Rocket health services in Uganda

PDIG-D-25-00072R2

Dear Dr. Ssenfuka,

We're pleased to inform you that your manuscript has been judged scientifically suitable for publication and will be formally accepted for publication once it meets all outstanding technical requirements.

Within one week, you'll receive an e-mail detailing the required amendments. When these have been addressed, you'll receive a formal acceptance letter and your manuscript will be scheduled for publication.

An invoice for payment will follow shortly after the formal acceptance. To ensure an efficient process, please log into Editorial Manager at https://www.editorialmanager.com/pdig/ click the 'Update My Information' link at the top of the page, and double check that your user information is up-to-date. For billing related questions, please contact billing support at https://plos.my.site.com/s/.

Kind regards,

Haleh Ayatollahi

Section Editor

PLOS Digital Health

Additional Editor Comments (optional):

Reviewer #1:

Reviewers' comments:

Reviewer's Responses to Questions

**Comments to the Author**

Reviewer #1: All comments have been addressed

publication criteria?

Reviewer #1: Yes

3. Has the statistical analysis been performed appropriately and rigorously?

Reviewer #1: Yes

4. Have the authors made all data underlying the findings in their manuscript fully available (please refer to the Data Availability Statement at the start of the manuscript PDF file)?

Reviewer #1: Yes

5. Is the manuscript presented in an intelligible fashion and written in standard English?

PLOS Digital Health does not copyedit accepted manuscripts, so the language in submitted articles must be clear, correct, and unambiguous. Any typographical or grammatical errors should be corrected at revision, so please note any specific errors here.

Reviewer #1: Yes

Reviewer #1: Accepted from my end

**Do you want your identity to be public for this peer review?** For information about this choice, including consent withdrawal, please see our Privacy Policy

Reviewer #1: Yes: Bala Nimmana
